# Increased Immunity against the Oral Germs *Porphyromonas gingivalis* and *Prevotella intermedia* in Different Categories of Juvenile Idiopathic Arthritis

**DOI:** 10.3390/biomedicines10102613

**Published:** 2022-10-18

**Authors:** Franck Zekre, Rolando Cimaz, Mireille Paul, Teresa Giani, Louis Waeckel, Anne-Emmanuelle Berger, Jean-Louis Stephan, Myriam Normand, Stéphane Paul, Hubert Marotte

**Affiliations:** 1LBTO Team, INSERM, 1059, SAINBIOSE (SAnté INgéniérie BIOlogie St-Etienne), Université Jean Monnet, 42055 Saint-Etienne, France; 2CIRI (Centre International de Recherche en Infectiologie), Équipe GIMAP (Team 15), INSERM U1111, CNRS, ENS, UCBL1, Université Jean Monnet, Université de Lyon, 42055 Saint-Etienne, France; 3Department of Pediatric, Hôpital Nord, University Hospital Saint-Etienne, 42055 Saint-Etienne, France; 4Department of Clinical Sciences and Community Health, University of Milano, 20122 Milano, Italy; 5Department of Medical Biotechnology, University of Siena, 53100 Siena, Italy; 6Laboratoire d’Immunologie, Hôpital Nord, University Hospital Saint-Etienne, 42055 Saint-Etienne, France; 7Department of Rheumatology Hôpital Nord, University Hospital Saint-Etienne, 42055 Saint-Etienne, France

**Keywords:** juvenile idiopathic arthritis, oligoarthritis, enthesis related arthritis, anti-*Porphyromonas gingivalis* antibody, anti-*Prevotella intermedia* antibody

## Abstract

(1) Background: The link between periodontal disease and rheumatoid arthritis (RA) is now widely reported. Several studies suggest the role of *Porphyromonas gingivalis* (*P. gingivalis*) in the pathophysiology of RA and some observations highlight the improvement of the disease activity induced by therapies against *P. gingivalis*. We have very little data on the prevalence of *P. gingivalis* carriage in patients with juvenile idiopathic arthritis (JIA) and its possible involvement in the pathophysiology of inflammatory joint diseases in children. (2) Methods: The specific IgG responses against *P. gingivalis* and *Prevotella intermedia* (*P. intermedia*) were determined in a cohort of 101 patients with JIA and 19 patients with other autoimmune diseases (inflammatory bowel disease and type 1 diabetes). (3) Results: Specific anti-*P. gingivalis* and anti-*P. intermedia* IgG titers were higher in JIA group than in control groups. These differences were mainly observed in the oligoarthritis group. The same pattern was observed in enthesitis-related arthritis (ERA). (4) Conclusions: Children with oligoarticular and ERA subsets had higher IgG titers to *P. gingivalis* and *P. intermedia*. These results suggest involvement of an oral dysbiosis in the occurrence of JIA in these subgroups.

## 1. Introduction

Rheumatoid arthritis (RA) is the most common inflammatory joint disease in adults. RA is characterized by the presence of an inflammatory infiltrate in the synovial tissue and by the presence of anti-citrullinated peptide antibodies (ACPA) in the blood [1]. Like RA, juvenile idiopathic arthritis (JIA) is characterized by systemic inflammatory and joint disease affecting children under 16 years of age [2]. The current classification groups the different forms of JIA into 7 distinct entities (systemic forms, polyarticular forms with or without rheumatoid factors, oligoarticular forms, enthesopathy-associated inflammatory arthritis (ERA), psoriasis-associated arthritis and unclassifiable arthritis) [3,4]. The exact etiology of JIA is still unknown. To date, the various hypotheses put forward on the occurrence of JIA integrate genetic and environmental frameworks. Recent data strongly suggest an involvement of periodontal diseases (PD) and in particular *Porphyromonas gingivalis* (*P. gingivalis*) in the occurrence of RA through the induction of ACPA [5,6,7]. *P. gingivalis* was mainly observed in case of severe PD. Several studies mention the beneficial effect of *P. gingivalis* treatment on disease activity in RA [8]. Recently, *P. gingivalis* infection has explained the occurrence of inflammatory arthritis in rodents [9]. However, there are very few studies on the prevalence of *P. gingivalis* in JIA patients and the putative involvement of the germ in the development of inflammatory joint disease in the pediatric population [10,11]. These studies highlight the presence of anti-*P. gingivalis* IgG in certain subgroups of JIA such as the polyarticular forms with the presence of ACPA and the ERA subsets. Several studies highlight the involvement of several microorganisms, including *Prevotella intermedia* (*P. intermedia*), in the pathophysiological mechanisms of chronic PD [12,13]. Moreover, *P. intermedia* has already been detected in the sera of RA patients [14]. Here, we determined the presence of specific IgG to *P. gingivalis* and *P. intermedia* in a cohort of JIA patients.

## 2. Materials and Methods

This is a European study, based on collaboration between France and Italy. Sera from 101 patients meeting the ILAR classification criteria for JIA [15] and followed in Italy (Milano) from 2018 to 2020 were used. Patient sera were collected at the time of disease diagnosis without any treatment being initiated. The median age of the JIA patients was 13 years. The study was approved by the ethics committee of Milano Area 2 (#690-2017), Italy, and informed consent was obtained from each subject and/or legal guardian. For the control group, sera were collected from patients with two other autoimmune diseases (type 1 diabetes and juvenile inflammatory bowel disease [IBD]) followed in Saint-Etienne (France) between 2013 and 2019 (IRBN1692021/CHUSTE). 

An anti-cyclic citrullinated peptides 2nd generation (anti-CCP2), and rheumatoid factor (RF, IgA and IgM) were measured by ELISA method on ImmunoCap 250 (Phadia, Thermo Fisher Scientific, Uppsala, Sweden). IgG antibodies against *P. gingivalis* and *P. intermedia* were performed as previously described [14]. Briefly, *P. gingivalis* strain ATCC 33277 and *P. intermedia* CIP 103607 were grown on sterile non-selective agar containing defibrinated sheep blood, supplemented with 0.0002% menadione sodium bisulfite and 0.4% hemin chloride in an anaerobic chamber for 7 days at 37 °C. *P. gingivalis* or *P. intermedia* solution was heated to 60 °C for 45 min, filtered (0.22 μm), diluted 1:10, coated on a 96-well plate and then incubated overnight at 4 °C. Plasma was diluted 1:900 in PBS containing 1% BSA and incubated (in duplicate) for 2 h at room temperature. Plates were washed as described above and incubated with peroxidase-conjugated goat anti-human IgG H+L (Jackson ImmunoResearch, West Grove, PA, USA) (diluted 1:50,000 in PBS) for 2 h at room temperature. After a final wash, detection was made by tetramethylbenzidine substrate (R&D Systems, Minneapolis, MN, USA). The reaction was stopped by the addition of H_2_SO_4_ (1M) solution and absorbances were measured at 450 nm. Cut-off values for seropositivity to *P. gingivalis* and *P. intermedia* were determined by concentrations higher than 95th percentile from a group of 27 healthy blood donors. 

A calibration curve was systematically done for each plate with dilutions of a pool of positive plasma diluted six times from 1:100 to 1:16,200 to correct for plate-to-plate variation. Two plasma controls (high and low positives) were included on all plates. Results are expressed in arbitrary units (AU) defined by the pool dilution (10 AU = 1:16,200 to 2430 AU = 1:100) [14]. 

Statistical analyses were performed using Prism from GraphPad Software (version 8). Statistical significance was assessed at *p* < 0.05. Antibody titers were represented by optical density values. Optical density values were compared between the different groups and the control group using non-parametric tests (Kruskal–Wallis one-way ANOVA, followed by Dunnett’s multiple comparison post hoc test). Correlation tests were performed on age and on anti-*P. gingivalis* and anti-*P. intermedia* concordance in the whole cohort. Correlation analysis was performed by non-parametric Spearman test.

## 3. Results

The study group comprised oligoarticular JIA (n = 48 cases; 47.5%), polyarticular JIA (n = 32 cases; 31.7% including 3 patients with RF detected), ERA (n = 9 cases; 8.9%) and systemic JIA (n = 12; 11,9%). The control group comprised 10 (40%) children who had type I diabetes and 15 (60%) who had inflammatory bowel disease (IBD). There was no statistical difference in age between the different groups (JIA vs. controls or between JIA subsets). Of the 101 JIA patients, 71% were female compared with 47% of the 19 control patients. The Kruskal–Wallis test showed a heterogeneous distribution of *P. gingivalis* and *P. intermedia* titers between the different groups. Specific *P. gingivalis* IgG titers were significantly elevated in the JIA cohort, particularly in the oligoarthritis vs. control subgroup (3 [2.2–3.3] vs. 2.4 [1.1–2.7]; *p* < 0.01; Figure 1a) and in the ERA vs. control group (3.3 [2.6–5.4] vs. 2.4 [1.1–2.7]; *p* < 0.01). Similar results were observed for the relative titers of anti-*P. intermedia* antibodies with high titers in oligoarthritis vs. controls (3.4 [2.4–4.7] vs. 2.7 [1.8–3.6]; *p* < 0.05; Figure 1b) and ERA vs. controls (4.7 [4–5.3] vs. 2.7 [1.8–3.6]; *p* < 0.01). Only 2 patients had a high anti-CCP2 titer and, respectively, 3 patients and 2 patients had high IgM and IgA RF titer in polyarticular JIA. The RF titer and the anti-CCP2 titer were not different according to various JIA subsets (Figure 1c–e).

Age at diagnosis and sampling was quite different between patients. Therefore, we examined the influence of age on *P. gingivalis* and *P. intermedia* serology titers. No correlation between the age of the patients and the serology titers was observed (Figure 3).

## 4. Discussion

Like many inflammatory disorders, the etiology of JIA is still unclear. Here we have highlighted environmental factors, including oral germs, which have recently been found to be involved in RA pathogenesis. The association between periodontal disease and JIA was recently confirmed by a large meta-analysis [12]. Furthermore, JIA children with anti-CCP positivity showed more symptoms of poor oral health and higher antibody titers to *P. gingivalis* than JIA children without anti-CCP [11], as observed in adult patients with RA [8]. In fact, risk factors of RA combine some genetic markers, mainly the shared epitope HLA-DRB1 [15] and some environmental factors. Among them, smoking (active or passive) is now well established as the main environmental risk factor. However, an association between RA and severe PD was also extensively reported [16]. In addition, immunity against *P. gingivalis* was mainly high in early RA patients with anti-CCP without smoking in the ESPOIR cohort [6]. This finding suggested that *P. gingivalis* was another way for smoking to induce anti-CCPproduction. Recently, oral exposure of *P. gingivalis*, a risk factor of RA, was confirmed by a meta-analysis of clinical data [17]. However, it still was only an association without a physiopathology link between *P. gingivalis* and RA induction. The answer came from an experimental animal model. In a rat, oral exposure of *P. gingivalis* was able to induce arthritis after an induction of PD and then including anti-CCP in the blood as observed in preclinical RA. After 8 months, a slight arthritis with bone damage was observed similarly to the damage observed in the rat adjuvant induced arthritis model at the time of the arthritis onset [9]. In another study, it was reported that in individuals at risk for RA, there was intestinal dysbiosis with an enrichment of *Prevotella copri* as compared with controls [18]. However, several studies investigated the effect of intensive PD therapy on RA activity without clear effect. For instance, in the French ESPOIR cohort of early RA, an interventional study was conducted. RA patients received general recommendations of good oral hygiene with teeth brushing, daily antiseptic mouthwash, and twice a year scaling according to the guidelines in the interventional group. In the control group, no recommendations were made. As expected, an improvement of the periodontal status in case of severe PD was observed. However, only a trend for a greater reduction of the DAS28 was observed in case of PD or presence of pathogenic bacterial load in the interventional group versus control [19]. Nevertheless, in the general population, a good dental hygiene is recommended. This is particularly the case in diseases with an increased risk of infection due to the disease itself and its treatment, such as RA. At present, there are no data on targeting periodontal disease in patients with pre-RP or RA with periodontal disease or a pathogenic oral bacterial load that would demonstrate a new way to prevent the onset of RA or improve the outcome of RA in the future.

In fact, our study described high immunization titers against two germs (*P. gingivalis* and *P. intermedia*) according to JIA subsets. The high level of specific anti-*P. gingivalis* IgG in the ERA form of JIA had already been reported [10]. Thus, we confirmed these data in a larger cohort and showed the presence of high specific anti-*P. gingivalis* IgG also in oligoarthritis, which constitutes the most common subgroup of JIA.

Unlike *P. gingivalis*, *P. intermedia* does not possess the enzymatic arsenal able to induce the citrullination of peptides observed in RA. However, *P. gingivalis* has been regularly used as a biomarker of chronic inflammation of the oral sphere observed in periodontal disease. The association observed in our study between *P. gingivalis* and *P. intermedia*, two germs of the oral sphere, reinforces the hypothesis of chronic oral inflammation in certain forms of JIA. All these observations corroborate the idea of the involvement of a dysbiosis as a trigger of these inflammatory joint pathologies [13,15] and could possibly allow other therapeutic approaches in the management of these patients, as suggested in RA patients [14].

However, our study has some weaknesses. First, the oral health status was not assessed in our patients. Second, no correlation between clinical parameters and our biological results was performed. Finally, like most studies, demonstration of an association between *P. gingivalis* or *P. intermedia* and JIA forms does not prove causality. Further large scale investigations into JIA could emphasize this apparent association and may prove the link between oral germs and JIA.

## 5. Conclusions

This was the first pediatric study to evaluate the correlation between *P. gingivalis* and *P. intermedia* in patients with JIA. JIA patients with oligoarthritis and ERA subsets have higher antibody responses to *P. gingivalis* suggesting involvement of periodontitis germs in various JIA subsets. The pathophysiological mechanisms of *P. gingivalis* involvement in the occurrence of these diseases remain to be determined. In the future, if the causality of *P. gingivalis* is demonstrated, this could imply regular dental follow-up for the secondary prevention of periodontal disease and could also allow the development of therapies directly targeting this germ in the management of these JIA subgroups.

## Figures and Tables

**Figure 1 biomedicines-10-02613-f001:**
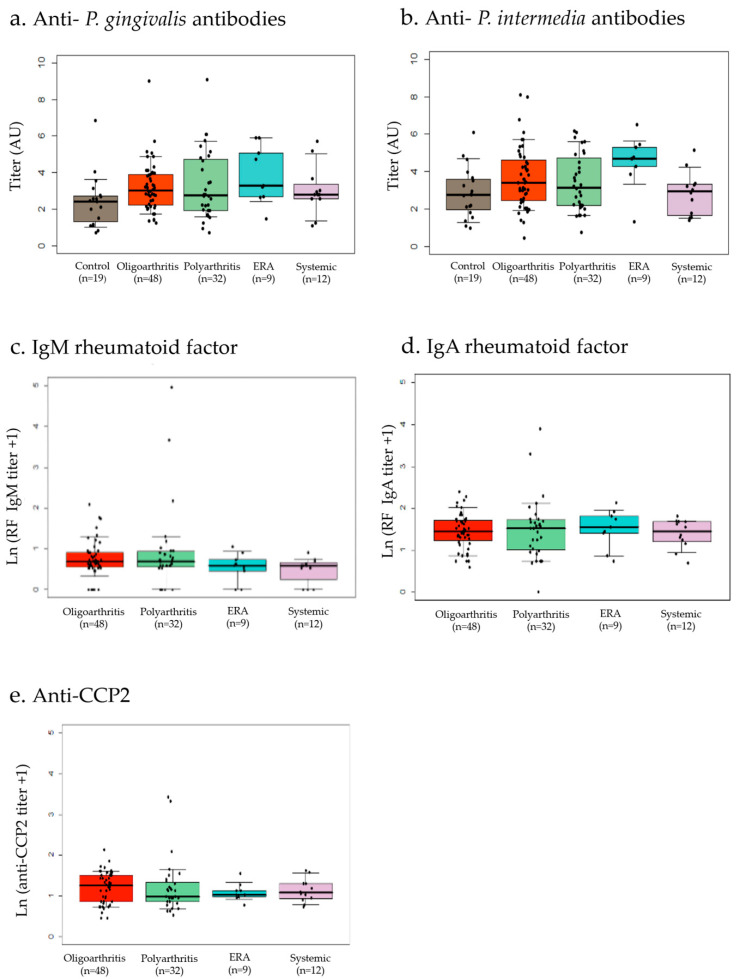
Heterogeneity of anti-*P. gingivalis* and anti-*P. intermedia* IgG titers according to juvenile idiopathic arthritis (JIA) subsets. The anti-*P. gingivalis* IgG titers are shown in panel a and the anti-*P. intermedia* IgG titers in panel b. Each dot represented each individual OD value. From the boxplot, the bar indicates the interquartile range and the median. Anti-*P. gingivalis* (**a**) and anti-*P. intermedia* (**b**) IgG values are significantly elevated in the oligoarthritis and enthesitis-related arthritis (ERA) subgroups compared to the control group. The IgM RF titer was not different according to various JIA subsets (**c**). The titer of FR IgA was similar in the different categories of JIA (**d**). The anti-citrullinated peptide titers were not different according to various JIA subsets (**e**). A strong positive correlation between anti-*P. gingivalis* and anti-*P. intermedia* antibody titers was observed (*r²* = 0.62; *p* = 0.01; Figure 2).

**Figure 2 biomedicines-10-02613-f002:**
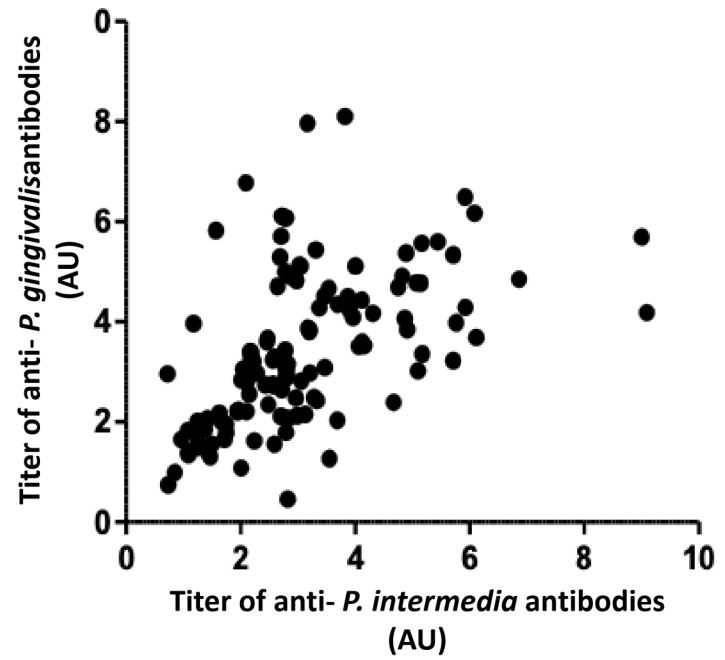
Strong correlation between titers of anti-*P. gingivalis* and anti-*P. intermedia* antibodies in the whole cohort (individuals with JIA and controls). Dots represent the values of *P. intermedia* versus the values of *P. gingivalis* for each individual. There is a strong correlation (*r²* = 0.62) between the values of *P. gingivalis* and *P. intermedia* (*p* < 0.05).

**Figure 3 biomedicines-10-02613-f003:**
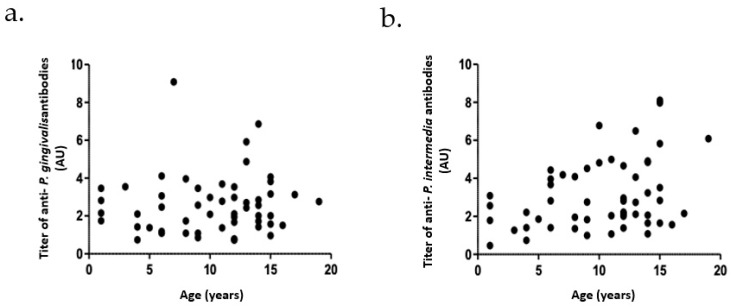
Lack of correlation between age of individuals and anti-*P. gingivalis* and anti-*P. intermedia* antibody levels. Dots represent the values of *P. gingivalis* and *P. intermedia* as a function of age. Panels (**a**,**b**) show a lack of correlation (*r*^2^ = 0.09 and *r*^2^ = 0.31, respectively) between the age of the patients and the anti-*P. gingivalis* and anti-*P. intermedia*.

## Data Availability

Data supporting the findings of this study are available within the article. Raw data can be obtained from the corresponding author (H.M.) upon reasonable request.

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
