# Peer review of "Increased Immunity against the Oral Germs Porphyromonas gingivalis and Prevotella intermedia in Different Categories of Juvenile Idiopathic Arthritis"

_biomedicines, 2022, doi:10.3390/biomedicines10102613_

Round 1

Reviewer 1 Report

Please find attached the comments.

Good job.

Regards

Author Response

Reviewer#1

In this paper, Franck Zekre et all, investigated the immune response against the oral germs Porphyromonas gingivalis and Prevotella intermedia in different categories of juvenile idiopathic arthritis (JIA).
Authors written an interesting communication, the experimental design, data analysis, and conclusions are appropriate for the study for the most part. However, the manuscript's impact can be further improved by expanding the points below.

We thank the reviewer for the positive comment of our manuscript.

Line 46: After the following sentence: “the exact etiology of JIA is still unknown. “Please explain the importance of IL-2 in the regulation of T cell functions in JIA. As you known, anti-inflammatory T regulatory cells (Tregs) are known for their essential role in the conservation of tolerance to self-antigens as well as in the regulatory mechanisms of immune-mediated inflammation. IL-2 is a pleiotropic cytokine indispensable for proper Treg cell function, making it an attractive target in autoimmune disease.

Thank you for this critical point. Our work focused on the role between immunological dysregulations and the occurrence of various JIA subtypes. However, the role of regulatory T and the involvement of IL2 and its receptor is quite far away from our work. So we still believe that adding our suggestion will be confusing in our work.

Line 56-59: “several studies..”: add references

We thank the reviewer for this comment. As suggested we added the followed references #12-13.

Line 63: Please explain which test did you use for determining the power of the research study and the result obtained by the analysis.

The test used to explore heterogeneity of serology according to the subset of AJI was the non-parametric test Kruskal-Wallis one-way ANOVA. To avoid the confusion, we added Kruskal Wallis” before one-way ANOVA in line 99.

Line 63: Please add references about the ILAR classification criteria for JIA.

We thank the reviewer for this comment. As suggested, we added the reference #15 in Line 65.

Line 62-64: “Patient sera were collected at the time of disease diagnosis without any treatment being initiated”. What is the nationality of the patients enrolled in the study? Are patients and controls French?

We thank the reviewer to point this confusing point. We confirmed that patients were at the beginning of the disease without treatment (line 66). JIA patients were enrolled in Milano (Italy) and controls in Saint-Etienne (France). To clarify we removed “France (Saint-Etienne) and” inline 65 and we added “(France)” in line 72.

Line 66: “The median age of the JIA patients was 13 years”. Is there a statistical difference in age and sex between 101 patients with JIA and 19 patients with other autoimmune diseases (inflammatory bowel disease and type 1 diabetes)?

We thank the reviewer for its comment. No significant difference for age between groups (JIA or controls and inside the various JIA subsets) were observed. So we added the following sentence “There was no statistical difference in age between the different groups (JIA vs. controls or between JIA subsets)” in Line 109.

Line 69: “For the control group, sera were collected from patients with two other autoimmune diseases (type 1 diabetes and juvenile inflammatory bowel disease 70 [IBD]) followed in Saint-Etienne between 2013 and 2019 (IRBN1692021/CHUSTE)”. Please perform the same experiment analysing the immune response against P. intermedia and P. gingivalis in heathy patient without autoimmune diseases but with periodontitis sorted by gender and age.

Thank you for your very pertinent remark. However, we were not able to provide this other control groups for several reasons. Firstly, periodontal disease in Western countries is quite rare in children and collecting blood will require a large-scale study with dentists. Secondly, it is difficult to organize blood collections in healthy children with huge difficulties to convince our ethical committee. We still believe that our proposed control group is still the best since they have another chronic inflammatory disease.

Line 100: Results. Please perform a correlation analysis between clinical parameters and biological results obtained.

Due to the small number of patients in each subset, it was difficult to perform this kind of correlation. Furthermore, by strongly increasing the number of statistical tests, we would increase the probability to have a significant result. So we did not add more data.

Line 122: Figure 1, please add the p-values obtained by ANOVA analysis and the cut off line in the graphs. Add also AU/mL in all graphs. Other than that please explain the outliers significance (see Fig 1 A, B, C, D, E).

Thank you for your comments.

The P-values are present in the text and we used the "Kruskal Wallis" ANOVA, followed by Dunnett’s multiple comparison post hoc test. We added this precision in Line 98.

We still believe that AU is the right unit.

Outliers correspond to heterogeneity of our results. We already observed some outlier in rheumatoid arthritis (ref#14).

Line 130: Figure 2, please explain the significance about correlation analysis between titers of anti-P. gingivalis and anti-P. intermedia antibodies. Please argue this result and draw conclusions, perhaps with the help of bibliographic data.

Thank you for your very pertinent remark which illustrates once again the interest between the microbiota, its alteration and the occurrence of dysimmune pathology. However, the significance of this correlation with the hypothesis that follows with the reference is illustrated in the discussion from line 198. This association between P.gingivalis and P. intermedia could reinforce the idea of oral dysbiosis in the triggering of JIAs because P.intermedia is also an important marker of periodontal pathologies.

Reviewer 2 Report

You have to do a thorough verification of your English language !

Author Response

Thank you for your positive review

Reviewer 3 Report

Please correct the errors beow:

Line

26:  juvenile arthritis idiopathic (JIA) – corr: „juvenile idiopathic arthritis (JIA)“

35:  Enthesis related arthritis;- corr „Enthesis -  related arthritis“

36:  Anti-Porphyromonas gingivalis antibody; Anti-Prevotella intermedia antibody – abbr. was used previously, it can be used also in this paragraph, corr-  „Anti- P. gingivalis antibody; Anti- P. intermedia antibody”

49: Porphyromonas gingivalis (P. gingivalis), corr: P. gingivalis

58: Prevotella intermedia (P. intermedia), corr P. intermedia

108: Kruskal Wallis, corr: „Kruskal- Wallis”

115: anti-CPP2, corr. „anti-CCP2”

156: periodontal disease (PD) – this abbr is mentioned before, corr: „PD”

187: citrunillation, corr „citrullination”

195: First the, corr: „First, the”

Author Response

We thank the reviewer for the positive comment of our manuscript.

Please see below the reply to comments.

Reviewer#3

Line 26:  juvenile arthritis idiopathic (JIA) – correctly: „juvenile idiopathic arthritis (JIA)“

Thank you to find this mistake. We updated by the right denomination.

Line 35:  Enthesis related arthritis;- corr „Enthesis - related arthritis“

Thank you to find this mistake. We updated by the right denomination.

Line 36:  Anti-Porphyromonas gingivalis antibody; Anti-Prevotella intermedia antibody – abbr. was used previously, it can be used also in this paragraph, corr-  „Anti- P. gingivalis antibody; Anti- P. intermedia antibody”

We do not agree. Abbreviation was defined in the abstract. Traditionally, abbreviations were not used in key words.

Line 49: Porphyromonas gingivalis (P. gingivalis), corr: P. gingivalis

Line 58: Prevotella intermedia (P. intermedia), corr P. intermedia

We do not agree. Abbreviation was defined in the abstract. Traditionally, it could be defined again in the main manuscript.

Line 108: Kruskal Wallis, corr: „Kruskal- Wallis”

Thank you to find this mistake. We updated by the right denomination.

Line 115: anti-CPP2, corr. „anti-CCP2”

Thank you to find this mistake. We updated by the right denomination.

Line 156: periodontal disease (PD) – this abbr is mentioned before, corr: „PD”

Thank you, we updated the abbreviation

Line 187: citrunillation, corr „citrullination”

Thank you to find this mistake. We updated by the right denomination.

Line 195: First the, corr: „First, the”

Thank you to find this mistake. We updated by the right denomination.

Round 2

Reviewer 1 Report

The authors answered the questions correctly. I believe the manuscript has been sufficiently improved to warrant publication in Biomedicines.

Kind regards